# *Helicobacter pylori*: A Contemporary Perspective on Pathogenesis, Diagnosis and Treatment Strategies

**DOI:** 10.3390/microorganisms12010222

**Published:** 2024-01-22

**Authors:** Asghar Ali, Khalid I. AlHussaini

**Affiliations:** 1Clinical Biochemistry Laboratory, Department of Biochemistry, School of Chemical and Life Sciences, Jamia Hamdard, New Delhi 110062, India; 2Department of Internal Medicine, College of Medicine, Imam Mohammad Ibn Saud Islamic University (IMSIU), Riyadh 4233-13317, Saudi Arabia

**Keywords:** pathogenesis, treatment strategies, virulence factors, triple therapy, antibiotic resistance, diagnostic tools

## Abstract

*Helicobacter pylori* (*H. pylori*) is a Gram-negative bacterium that colonizes the gastric mucosa and is associated with various gastrointestinal disorders. *H. pylori* is a pervasive pathogen, infecting nearly 50% of the world’s population, and presents a substantial concern due to its link with gastric cancer, ranking as the third most common cause of global cancer-related mortality. This review article provides an updated and comprehensive overview of the current understanding of *H. pylori* infection, focusing on its pathogenesis, diagnosis, and treatment strategies. The intricate mechanisms underlying its pathogenesis, including the virulence factors and host interactions, are discussed in detail. The diagnostic methods, ranging from the traditional techniques to the advanced molecular approaches, are explored, highlighting their strengths and limitations. The evolving landscape of treatment strategies, including antibiotic regimens and emerging therapeutic approaches, is thoroughly examined. Through a critical synthesis of the recent research findings, this article offers valuable insights into the contemporary knowledge of *Helicobacter pylori* infection, guiding both clinicians and researchers toward effective management and future directions in combating this global health challenge.

## 1. Introduction

*Helicobacter pylori*, a ubiquitous, microaerophilic, spiral-shaped, Gram-negative bacterium residing in the human stomach, has profoundly impacted the landscape of gastroenterology and infectious diseases since its discovery in 1982 [1]. Before its identification, peptic ulcer disease was primarily attributed to stress and dietary factors, fundamentally altering our understanding of its etiology. This revolutionary discovery earned Barry J. Marshall and Robin Warren the Nobel Prize in Physiology or Medicine in 2005 and laid the foundation for a comprehensive investigation into *H. pylori*’s role in various gastrointestinal disorders [2]. As the scientific community has continued to unravel the complexities of this pathogen, the field of *H. pylori* research has witnessed remarkable advancements. *H. pylori* colonizes the gastric mucosa of approximately half the world’s population, making it one of the most prevalent human infections [2]. Its ability to establish a persistent and often lifelong infection in the stomach lining has earned it a reputation as a formidable pathogen. The bacterium’s remarkable adaptability to the acidic and inhospitable gastric environment has led to a plethora of host responses and pathological outcomes [3,4]. From its initial association with peptic ulcers, our understanding has expanded to include its role in gastritis, duodenal ulcers, gastric cancer, mucoid-associated lymphoid tissue (MALT) lymphoma, and a range of extra-gastric conditions, including neurological, ocular, hematologic, cardiovascular, and dermatological diseases, which afflict millions worldwide and have substantial economic and healthcare burdens [4,5,6]. The complex interplay between *H. pylori* and its human host have spurred intensive research efforts to elucidate its pathogenesis, develop accurate diagnostic methods, and refine the treatment strategies. The significance of *H. pylori* lies not only in its prevalence, but also in its wide-ranging impact on human health. Additionally, *H. pylori* is recognized as a Group 1 carcinogen by the International Agency for Research on Cancer (IARC); it is the only bacterium on the list with a strong association with gastric adenocarcinoma and is one of the leading causes of cancer-related deaths globally [7]. Furthermore, the bacterium’s implications extend beyond the stomach, with the links to autoimmune disorders, cardiovascular diseases, and metabolic syndromes being explored. Understanding the pathogenesis, accurate diagnosis, and effective management of *H. pylori* infections are paramount in mitigating their impact on public health.

### 1.1. H. pylori Infection: The Global Scenario

*H. pylori* infection is quite common worldwide, with approximately 50% of the world’s population reported to be infected [8]. The prevalence, however, is observed to vary depending on the geographical conditions, people’s socioeconomic status, temperature, and the hygiene conditions in a particular region [9]. Despite the variation, a trend of increased occurrence of this infection has been observed in the case of developing countries with a low socio-economic status [10,11], with a particularly high incidence of gastric cancer as well. According to an extensive meta-analysis conducted by Li et al., high-income countries and countries that have appreciable health coverage have a relatively lower rate of occurrence of *H. pylori* infections [12]. Another report by Hooi et al. revealed a higher incidence of H. pylori infections in African countries (more than 70%), while the lowest prevalence was recorded in Oceania (less than 25%) [13]. Despite the rising cases of antibiotic resistance, a significant reduction has been observed in global *H. pylori* infections since 1980 in Africa, the West Pacific region, and South East Asia, with a similar decrease in cases of gastric cancers associated with *H. pylori* as well [14,15]. Additionally, the prevalence patterns of the infection also vary within different regions of the same country [16].

### 1.2. Significance of Study

*H. pylori* is highly correlated with duodenal ulcers (in up to 90% of cases), gastric ulcers (in approximately 80%), and malignancies, contributing to conditions such as mucosa-associated lymphoid tissue (MALT) lymphoma and gastric cancer (seen in as many as 90% of cases) [4,17,18]. In 2014, the World Health Organization (WHO) advocated for the eradication of *H. pylori* to reduce the global gastric cancer-causing mortality rate, and in 2017, it identified clarithromycin-resistant *H. pylori* strains as a significant public health threat. This review provides an overview of *H. pylori*’s disease characteristics, encompassing epidemiology and clinical presentation, and discusses the latest advances in evaluating and managing this infection. A timeline of significant events related to *H. pylori* discovery and the management of infection is represented in Figure 1.

This review, endeavors to synthesize the wealth of knowledge accumulated in the field of *H. pylori* research. Our primary objectives are to provide a comprehensive overview of the bacterium’s pathogenesis, including the molecular mechanisms behind its infection, its role in triggering inflammatory responses, and its associations with gastrointestinal diseases. We will also explore the latest advancements in diagnostic techniques, emphasizing the importance of accuracy and efficiency in detection. In the context of a changing landscape of antibiotic resistance, we will assess various treatment strategies, including antibiotic therapy, proton pump inhibitors, and emerging therapeutic options. Moreover, we will highlight the concept of personalized medicine as a promising approach to combat *H. pylori* infections. By offering critical insights into *H. pylori*’s contemporary landscape and future directions in its management, this review aims to serve as a comprehensive resource for clinicians, researchers, and healthcare professionals engaged in the fight against this clinically significant pathogen.

## 2. Pathogenesis of *Helicobacter pylori*

The pathogenesis of *H. pylori* may be studied at three distinct stages: the attachment to and colonization of the gastric mucosa, triggering and evading the immune responses of the host, and finally, the establishment of the disease [19].

### 2.1. Dispersal and Routes of Infection

*H. pylori* infections are often asymptomatic and usually acquired during childhood. However, around 30% of infected persons may show signs of gastrointestinal diseases, such as mild-to-severe cases of peptic ulcers, gastritis, and even gastric cancer and MALT lymphoma. *H. pylori* infections have a narrow host range, and therefore, the transmission routes usually include vertical and horizontal transmission. The former includes person-to-person transmission between genetically related individuals, while the latter includes persons exposed to infected people of a similar socioeconomic status [20]. While the exact route of transmission remains unclear, the four major routes of *H. pylori* infection include the fecal–oral route, the oral–oral route, the gastric–oral route, and gastro-gastric route which may occur through the ingestion of contaminated food, water or during endoscopic procedures [21].

### 2.2. Molecular Mechanisms of Infection

#### 2.2.1. Attachment and Colonization

The molecular mechanisms orchestrating *H. pylori* infections are sophisticated processes crucial for the bacterium’s successful establishment in the gastric mucosa. At the forefront of infection initiation is *H. pylori*’s adhesion and colonization strategies. The colonization of the pathogen is initiated first by the chemotaxis of the bacterial cells to the target site, which is mediated by the presence of certain receptors present on the host cells, mainly belonging to the Tlp family [22]. These receptors are known to be triggered by the presence of chemical signals, including urea, lactic acid, ROS species, and gastric juice, facilitating chemotactic responses to the gastric epithelium [22]. Flagellar motility allows *H. pylori* to navigate through the mucous layer and reach the gastric epithelium, contributing to its ability to firmly adhere to and colonize the mucosa [23]. Additionally, *H. pylori* can form biofilms, structured bacterial communities embedded in an extracellular matrix, enhancing adherence and providing protection against the host defenses and treatment therapies [24]. Furthermore, the bacterial outer membrane proteins, aka the surface adhesins, including BabA, SabA, AlpA/B, and OipA, enable the bacteria to adhere to gastric epithelial cells, paving the way for its persistence within the stomach [25].

#### 2.2.2. Production of Virulence Factors

Various virulence factors, including those involved in motility, adhesion, urease and cytotoxin production, are essential for the pathogenesis of this bacterium [26].

#### 2.2.3. CagA and VacA

Upon attachment, *H. pylori* engages in intricate host interactions, deploying an arsenal of virulence factors that manipulate host cell signaling. Among these, the CagA (cytotoxin-associated gene A) pathogenicity island is the main orchestrator [27]. Injected into the host cells via a type IV secretion system (T4SS), CagA triggers a cascade of events, disrupting cellular functions and contributing to the development of gastric pathologies [28]. CagA undergoes tyrosine phosphorylation, leading to the activation of various cellular signaling pathways [29]. This includes the aberrant activation of the mitogen-activated protein kinase (MAPK) and phosphatidylinositol 3-kinase (PI3K) pathways [30,31]. The dysregulation of these pathways contributes to cellular morphological changes, the disruption of cell polarity, and the initiation of oncogenic processes. This molecular hijacking of host cell signaling is a hallmark of *H. pylori*-induced pathogenesis [32]. The vacuolating toxin, VacA, is another key virulence factor, that exhibits multifaceted effects on the host cells [33]. As the name suggests, VacA disrupts the integrity of the gastric epithelial barrier by forming channels or pores in the cell membrane, leading to increased permeability [26]. Within the host cells, VacA modulates the apoptotic pathways, leading to both pro- and antiapoptotic effects depending on the cell type and environmental conditions. Furthermore, VacA toxin contributes to the formation of vacuoles within the host cells, impacting the cellular structure and function [34]. This toxin also interferes with and evades the immune responses by affecting the function of T cells and other immune cells. The consequences of these molecular interactions extend to the host’s immune response [35,36]. *H. pylori* induces a chronic inflammatory state marked by the release of multiple pro-inflammatory cytokines, such as IL-8, IL-1α. IL-1β, and TNF α, among others, and the recruitment of immune cells to the gastric mucosa [37]. This inflammatory milieu is closely associated with the development of peptic ulcers, as the delicate balance between the protective mucosal mechanisms and bacterial aggression is disrupted [38]. The CagA and VacA proteins work together to induce *H. pylori*-associated gastric cancer. A study conducted by Abdullah et al. (2019) showed the functional interplay between the two oncoproteins, whereby the absence of VacA aids the host’s system is able to degrade CagA, thereby preventing the accumulation of CagA in the gastric epithelial cells [39]. Importantly, the long-term consequences of *H. pylori* infection include an increased risk of developing gastric cancer. Chronic inflammation, coupled with the release of genotoxins and the induction of genetic instability, creates an environment conducive to oncogenesis and other gastric malignancies [40]. The interplay between *H. pylori’s* virulence factors, such as CagA, VacA, and OipA, host genetic susceptibility, and environmental factors contribute to the complexity of this association [41]. Understanding these intricate molecular mechanisms is paramount for developing targeted therapeutic approaches and preventive strategies against *H. pylori* infections. The advances in elucidating these processes provide critical insights into the pathogenesis of *H. pylori*-related diseases and inform the development of novel interventions to mitigate the associated health risks.

#### 2.2.4. Urease Production and Survival at Low pH

Urease is one of the more abundantly produced proteins expressed by the pathogen, accounting for almost 15% of the total proteins of the bacterium [42]. The production of urease is a characteristic feature of *H. pylori* and is widely used in the diagnosis of the infection as well [43]. Several studies in this regard have proved the role of urease in enabling the survival of the bacterium in the extremely low-pH acidic environment of the stomach by breaking down the urea into ammonia and carbon dioxide, thereby forming a pH-neutral environment around the bacterial cells (Figure 2). The other enzymes, including carbonic anhydrase, arginase, glutamine synthase, glutamate dehydrogenase, and glutaminase, are also involved in the urease-dependent mechanism of survival under acidic conditions [44]. The hydrolysis of urea into ammonia also provides the pathogen with a steady source of nitrogen [45,46]. Survival at low pH is further supported by other urease-independent physiological factors, including the flagellar motility across the gastric mucus layer, which facilitates the movement of the bacterial cells in low-pH-level conditions [47]. Furthermore, DNA repair proteins, such as RecA, RecN, RecO, Hup, etc., are actively involved in repairing the DNA that may be damaged after being subjected to acid stress in the stomach [48,49]. *H. pylori* also exhibits chemotactic activity towards certain chemoattractants, such as carbonates and urea, which attract the bacterial cells towards the regions of higher pH in the gastric lining. Among the chemoreceptors of *H. pylori*, the Tlp family is crucial for the desirable chemotactic activity that promotes survival in an acid stress environment [50].

### 2.3. Immune System Modulation and Induction of Inflammatory Responses

The immune response to *H. pylori* infection is a dynamic interplay between the bacterial factors and the host’s immune system [51]. The pathogenesis of *H. pylori* infections unfold through a complex and multifaceted localized gastric inflammatory response [52]. In the innate immune phase, *H. pylori* induces a chronic inflammatory response in the gastric mucosa, mediated by the lipopolysaccharides and the peptidoglycan of the cell wall, and characterized by the release of neutrophils, macrophages, lymphocytes, and pro-inflammatory cytokines (such as interleukin-1β, interleukin-6, and tumor necrosis factor-α) and the recruitment of immune cells [53,54]. Innate immune activation spurred by bacterial components, like lipopolysaccharide and peptidoglycan, further amplifies the inflammatory cascade through pattern recognition receptors such as Toll-like receptors [55]. This sets the stage for the adaptive immune response, where the CD4+ T-helper cells play a crucial role, particularly in promoting a Th1 response characterized by interferon-gamma secretion. Regulatory T cells are recruited, contributing to immune suppression, while the B cells produce antibodies against *H. pylori* [56,57]. However, the bacterium employs immune evasion strategies, such as urease production, antigenic variation, inhibition of recognition by PRRs, and molecular mimicry, to subvert recognition and signaling by the T and B cells [58]). The sustained presence of *H. pylori* leads to chronic inflammation, causing ongoing tissue damage and remodeling. The gastric epithelial cells undergo changes in turnover, with increased proliferation contributing to mucosal damage and ulcer formation [59]. Concurrently, disruptions in the tight junctions compromise the mucosal barrier integrity, facilitating *H. pylori* infiltration into the deeper mucosal layers [60]. The sustained release of inflammatory mediators, coupled with the induction of genetic instability, creates an environment conducive to oncogenesis [61]. This sustained inflammation is a central component of *H. pylori*-associated diseases and is implicated in the development of peptic ulcers and gastric cancer [2]. However, in the neighboring non-infected gastric cells, an adaptive immune response is triggered which prompts increased survival and proliferation, ultimately leading to the development of precancerous lesions in the gastric epithelium [62].

### 2.4. Modulation of Mucin Production

*H. pylori* influences mucin production in the gastric mucosa, impacting the protective mucous layer. The carbohydrate component of the mucins act as ligands that enable the binding of the bacterium to the gastric mucosal lining [63]. The bacterium can alter the expression of mucin genes and interact with mucins, like MUC5A and MUC1, directly, resulting in inhibition or the impairment of mucin turnover [64]. The changes in mucin production influence the composition of the mucosal glycocalyx, affecting the adherence and colonization of *H. pylori*. Moreover, this modulation of mucin production contributes to the bacterium’s ability to evade the host defenses and establish persistent infections [65]. Unraveling the nuances of immune system modulation in *H. pylori* infection is paramount for devising effective therapeutic interventions and preventive strategies in the ongoing pursuit of managing the associated diseases. The ongoing research continues to unveil the complexities of this host–pathogen interaction.

## 3. Disease Associations

*H. pylori* mainly infects the mucosal layer of the stomach, owing to its ability to adapt to the acidic environment by neutralizing the local environment, owing to the production of urease and other factors [66]. Although the exact mode of transmission remains mostly unclear, it is usually transmitted through the oral–fecal route, or the gastric–oral route, or oral–oral route, and may spread through the ingestion of contaminated food and water. The tendencies for vertical or parental transfer have also been reported [67,68]. This infection is usually acquired during childhood, and only about 20–30% of the infected people actually show the symptoms of infection, i.e., the infection is widely asymptomatic [20,67]. As a result, undiagnosed infections may lead to chronic inflammations, including non-atrophic and eventually, atrophic gastritis, i.e., the inflammation of the gastric mucosa. Such chronic gastritis has often been linked with the development of gastric cancer, including adenocarcinomas and MALT lymphomas [69,70]. Approximately 10% of the infected individuals may develop peptic ulcers, while gastric adenocarcinoma is reportedly found to occur in almost 3% of the infected patients. *H. pylori* is also commonly associated with a number of gastric issues, including dyspepsia, which is commonly characterized by pain or discomfort in the abdomen [71]. The studies in this regard have frequently pointed towards a link between *H. pylori* infection and dyspepsia, as eradication-based studies have clearly demonstrated a reduction in functional dyspeptic symptoms [3,72]. *H. pylori* has frequently been credited as one of two the major risk factors (the other being NSAIDs) for the development of peptic ulcer disease (PUD), which is characterized by the formation of gastric lesions in the duodenum and the stomach due to penetration of the muscular mucosa [73,74]. Gastric cancer is reported to be the third most common type of cancer globally [75]. *H. pylori* is notoriously well-recognized for its involvement in the genesis of gastric cancer. In fact, more than 90% cases of all non-cardia gastric cancer cases are caused by *H. pylori* infection [69]. Various metanalytical studies in this regard have confirmed the correlation between *H. pylori* infection and the development of noncardia GC [76]. The CagA cytotoxin discussed in the previous section is considered to be the first bacterial oncoprotein integrally associated with the development of gastric carcinoma among infected individuals [77]. A range of host factors, including interleukins (IL-β) and the TNF, are known to increase the risk of noncardia GC [78]. In general, the beta-catenin signaling pathway has been linked to the onset of carcinogenesis post-infection by *H. pylori* [79]. The inflammation of the gastric epithelium and other factors, including cyclooxygenase, prostaglandins, disrupted cadherin–catenin interactions, etc., are reported to play a role in the development of gastric adenocarcinoma [80].

## 4. Diagnosis of *Helicobacter pylori* Infections

Most cases of *H. pylori* infections are asymptomatic; therefore, they often go unrecognized, leading to the development of chronic gastritis, which may further lead to the development of gastric cancer [81,82]. Additionally, *H. pylori* infection may manifest itself in the form of mild dyspepsia, and occasionally, more serious conditions, such as intestinal metaplasia, peptic ulcers, and MALT lymphomas [83]. The American College of Gastroenterology recommends testing for *H. pylori* in cases of PUD, MALT lymphomas, dyspepsia, and gastric cancer [84]. The alarm symptoms usually include vomiting, GI bleeding, dysphagia, and weight loss, among others [85]. The generally accepted diagnosis method involves a test-and-treat approach, and various invasive and non-invasive tests are available to detect *H. pylori* infections; the choice depends upon the patient’s age and their symptoms [86]. The invasive tests involve procedures, such as endoscopy, histology, rapid urease test, and cultural tests, while the non-invasive tests include breath tests and fecal tests, among others [87]. Although on the whole, most experts are of the opinion that no single test can be considered the gold standard for the diagnosis of *H. pylori*-mediated gastritis, certain tests are considered more reliable and preferable to others. These include the urea breath test (UBT) and histological examinations post biopsy [88]. While making a choice of the detection method, the clear analysis of the intent, the specificity, sensitivity, availability, costs, and limitations must be made [89]. Keeping in mind the individual shortcomings of each of the invasive and non-invasive examinations, a combination-based approach, such as RUT combined with histological examination and bacterial culture, for the detection of *H. pylori* infection may be a more suitable approach for an accurate diagnosis [90]. Such an effective diagnosis and eradication can further prevent the progression of gastric carcinogenesis and also has the potential to reverse unregulated cell growth [91].

### 4.1. Non-Invasive Tests

#### 4.1.1. Serological Assays

Serological tests are considered to be the more accurate of the non-invasive methods to diagnose a suspected case of *H. pylori* infection. These tests are based on the detection of pathogen-specific antibodies in bodily fluids, such as sera, saliva, or urine samples [92]. Serological examinations rely on the presence of anti-*H. pylori* IgG antibodies, which are commonly employed for the accurate diagnosis of suspected infections. Additionally, the other antigenic proteins, including CagA, Omp, UreA, and VacA, are effective candidates to confirm the infection. The advantages of using serological tests for the detection of *H. pylori* infection includes the accuracy of the tests and that such tests are unaffected by the use of PPI and antibiotics, which can often cause false negative results in the case of other diagnostic tests [93]. However, serology-based detection is not reliable for the confirmation of eradication, as the *H. pylori*-specific antibodies may persist even after eradication [6].

#### 4.1.2. Urea Breath Test (UBT)

The 13C urea breath test (UBT) relies on the characteristic production of ammonia and carbon dioxide following the hydrolysis of urea by the enzyme urease [88]. The fact that urease is not produced by mammalian cells is suitably considered a sign of *H. pylori* infection, as this microbe can effectively use urease to degrade the urea [94]. Essentially this test involves a C13 or C14 isotope-containing urea substrate, which is ingested by the patient. Exhalation samples of the patient, both before and after ingestion, are collected to detect the infection [95]. The use of radio-labeled (14C-UBT) or non-radiolabeled (13C-UBT) carbon isotopes in the urea substrate helps in the detection of labeled CO_2_ after exhalation using analytical techniques like mass spectrometry, indicating the presence of the pathogenic bacterium [95]. The preparation for this test involves discontinuation of antibiotics and intake of bismuth compounds and other such drugs, which may reduce the activity of bacterial urease, thereby avoiding the chance of acquiring a false negative result. The UBT is a reliable and highly sensitive and specific test, with a sensitivity above 95% and a specificity above 93% [96]. Given its principle, easy procedure and reliability, this test is suitable in the absence of alarm symptoms (weight loss, bleeding, an abdominal mass, and dysphagia—VBAD), for both the initial diagnosis and the post-treatment tests [86,97]. However, certain questions have been raised regarding its reliability, as urease may be produced by other gut-inhabiting bacteria as well (Table 1) [88].

#### 4.1.3. Stool Antigen Test (SAT)

The stool antigen test (SAT) or the fecal antigen test (FAT) is another reliable non-invasive test that is commonly employed to detect the presence of *H. pylori* in the fecal matter of suspected patients [98]. There are two types of SAT based on the choice of the analytical technique employed after stool collection, namely, enzyme immunoassay (EIA), while the other is based on immunochromatography (ICA) [99]. According to a study conducted by Gisbert et al. (2006), the accuracy of the monoclonal SAT was found to be higher than that of the polyclonal SAT, especially for the confirmation of post-treatment eradication efficiency [100].

### 4.2. Invasive Tests

Invasive tests for *H. pylori* detection include the collection of specimens via endoscopic biopsy procedures, which are followed by histological examinations, cultural approaches, the rapid urease test, and PCR-based analyses, all of which are carried out after a biopsy procedure [101]. Esophagogastroduodenoscopy (EGD) is, by and large, the most reliable and the most recommended diagnostic procedure for the accurate confirmation of suspected *H. pylori* infection, and is considered the gold-standard diagnostic procedure for the infection [102]. This test may be followed by other confirmations, including immunohistochemical procedures, cultural tests, and molecular methods, including PCR-based analysis [95].

#### 4.2.1. Histology

The histopathological examination of people suspected of infection by *H. pylori* is considered among the most reliable and specific diagnostic procedure, offering advantageous information about the extent of inflammation and associated pathologies, such as gastric cancer, intestinal metaplasia, and gastritis [103]. The staining method usually involves employed hematoxylin and eosin (H&E), although ancillary tests involving Geimsa, Genta stain, or Warthin–Starry silver stain and immunohistochemistry are used for accurate histological diagnosis, especially when H&E staining provides insufficient results under the conditions of the low density of the bacterium in the sample [103,104]. Immunohistochemistry (IHC) is a precise and highly sensitive and specific method for detecting *H. pylori* in gastric biopsies [103]. Histological diagnosis is affected by various factors, including the site, size, and number of biopsy specimens taken. The Sydney system recommends extracting biopsy specimens from five sites, including the pylorus, antrum, corpus, and angularis, to ensure optimal detection [105]. Histological examinations remain the most available, accurate, and cost-effective method, offering more sensitivity than the non-invasive and most of the invasive methods, and they are the more commonly preferred diagnostic choice when confirming suspected cases of *H. pylori* infection [106].

#### 4.2.2. Culture Examinations

In accordance with Koch’s postulates, microbiological examinations based on the in vitro culturing of the bacterium from biopsy specimens obtained after an endoscopy are also viable options for the accurate detection of *H. pylori*. However, certain disadvantages are associated with this test, including the fastidious nature of the microbe, the requirement of specific growth conditions and microbiological equipment, and low recovery rates of the bacterium from the infected samples [88]. The critical reviews in this regard deem the method to be invasive, time-expensive, and have a low sensitivity, and therefore, the other methods may be preferable over culture-based diagnosis [107]. However, the fact remains that culture examination is the only test that allows for the specific analysis of the microbiological characteristics of the strain type, and also determines the antibiotic sensitivity [108].

**Table 1 microorganisms-12-00222-t001:** Comparison of different techniques available for diagnosis of *H. pylori*.

Diagnostic Test	Sensitivity	Specificity	Advantages/Disadvantages	References
Urea breath test (UBT)	>95%	>95%	Advantages: Gold standard in many clinical diagnoses, cost-effective, reliable, simple, non-invasive, and can be used for confirming eradication of infection. Disadvantages: May give false negatives in the presence of other urease-producing bacteria and *Helicobacter* species. Low accuracy under conditions of gastritis and gastric malignancies, requires a high load of bacteria in the specimen. Requires expensive equipment.	[96,109]
Stool antigen test (SAT)	96%	97%	Advantages: Cost-effective, simple, rapid, and does not require expensive instruments. Disadvantages: It may give false negatives under low bacterial count, and accuracy is affected by recent intake of PPI and CAM; it is not useful for post-eradication confirmation.	[99,100]
Serological tests	85%	>80%	Advantages: Inexpensive and can be employed for patients who have recently undergone triple therapy. Only tests not affected by PPI intake or use of antibiotics. Disadvantages: Unreliable for ongoing infections and cannot be used to confirm eradication.	[87]
Rapid urease test (RUT)	80–90%	93–100%	Advantages: Rapid, inexpensive, and simple. Disadvantages: Invasive, requires additional confirmatory tests, and accuracy affected by intake of PPI and antibiotics.	[61]
Culture	70–90%	100%	Advantages: Gold standard for confirmation and can be used to ascertain antibiotic sensitivity. Disadvantages: Elaborate, time-consuming, expensive, and requires specific expertise in microbiology.	[110]
Histopathology	>95%	99%	Advantages: Gold standard in routine clinical diagnostics, provides additional information about associated pathologies, and extremely sensitive and specific.	[101]
Molecular methods (PCR)	96%	98%	Advantage: Sensitive even at very low bacterial counts. Disadvantages: Expensive, requires sophisticated equipment, and may give false positive results.	[88]

#### 4.2.3. Molecular and Genetic Markers

The recent advances in molecular biology have provided new avenues for detecting *H. pylori*. Given their accuracy and reliability, molecular diagnostic procedures should be preferred over the other methods discussed above. Methods such as RT-PCR allow for not only the detection of the bacteria, but also their possible resistance against the recommended antibiotics for the eradication of the infection [111]. The other approaches rely on the use of biomolecules as markers, enabling the early diagnosis of *H. pylori* infection using such biosensors. These markers include 16S rRNA, 16S rDNA, cagA, vacA, and the other factors associated with *H. pylori* [106]. In a study conducted in 2021, 16S rRNA amplicon sequencing was proposed as a suitable and accurate method for the detection of *H. pylori* infection using gastric biopsy specimens. Additionally, this method was especially advantageous in confirming the presence of the pathogen in cases with a negative test result using routine test methods [112]. In a recently conducted study, a novel RPA-CRISPR-Cas12a-based method was developed for the detection of virulence factors CagA and VacA and the 16S rDNA gene as well. This technique was found to be very sensitive (2 ng/microliter), and therefore, may be applied for the rapid, accurate, and sensitive detection of *H. pylori* in various settings (Table 1) [91].

## 5. Treatment Strategies

### 5.1. Antibiotic Therapy and Selection of Antibiotics

Currently, the widely accepted standard treatment strategy involves a triple therapy approach, whereby at least two antibiotics (usually amoxicillin and clarithromycin) and one proton pump inhibitor (PPI) are administered to the infected individual [113]. Antimicrobials, which are recommended in the Maastricht IV/Florence consensus report, when used in combination exert a triple mode of action, affecting the cell wall and bacterial protein synthesis by binding to the 50S ribosomal unit and affecting DNA integrity [6,114]. However, such an antibiotic-based treatment strategy is impeded by the prevalence of antibiotic resistance to the primary or the secondary antibiotic, the possible occurrence of problematic side effects, mucosal drug concentrations, and the expensive nature of such drugs [115,116]. In the case of the failure of the first regimen of treatment, a triple treatment comprising Levofloxacin is recommended [117]. However, the fact remains that the growing prevalence of antibiotic resistance among various clinical and non-clinical bacteria reduces the effectiveness of the treatment strategies that rely on antibiotics. A second-line therapy in cases of the failure of the first-line regimen includes a bismuth quadruple therapy (BQT) approach, comprising a bismuth salt in addition to a proton pump inhibitor, metronidazole, and tetracycline [118,119]. BQT may also be the treatment of choice under the condition of clarithromycin resistance [118]. In terms of the effectiveness of the treatment, a study conducted by Luther et al. (2010) revealed almost equivalent results, with similar outcomes for patient compliance, side effects, and eradication rates [120]. According to the Maastricht V/Florence consensus report, a concomitant fluoroquinolone–amoxicillin triple/quadruple therapy may also be employed as the second-line treatment option, as an alternative to BQT [121]. Alternatively, a concomitant therapy, a type of non-bismuth quadruple therapy involving the concomitant administration of a PPI, amoxicillin, clarithromycin, and nitrimidazole, has also been identified as an effective treatment option against triple therapy-resistant infections [103,122].

### 5.2. Proton Pump Inhibitors

PPIs are a group of chemical compounds that inhibit the gastric acid pump (H+/K+-ATPase) by binding to the cysteine residues of the proton pumps [123,124]. Some examples of proton pump inhibitors include pantoprazole, rabeprazole, omeprazole, tenatoprazole, etc. First introduced in 1989, such bezimadazole-derived acid suppressors are excellent healers of peptic ulcers and are given in combination with antibiotics in the treatment and eradication of *H. pylori* infections, which lead to exaggerated acid production in the infected individuals [125,126]. The administration of PPI in the eradication of *H. pylori* provides the benefit of increasing the bioavailability of the acid-susceptible antibiotics, and also having a direct effect on *H. pylori’s* growth and metabolism [127]. Although currently indispensable in the treatment of *H. pylori* infections, certain concerns have been raised regarding a few mild side effects, including an increased risk of infections by enteric bacteria, such as *C. difficile* and *Salmonella*, acute kidney injury, interference with calcium and magnesium homeostasis, and probably, an increased risk developing of coronary diseases [126].

### 5.3. The Rising Issue of Antibiotic Resistance

The now highly prevalent occurrence of antibiotic resistance has continued to plague the treatment strategies for various infectious diseases, and *H. pylori* infections are no different. Over the years, a higher incidence of treatment failures has been reported on a global scale [128]. The rates of resistance against the conventionally employed antibiotics are widely variable; i.e., cases of heteroresistance depending upon the region and the antibiotic class with considerable variation in each region have been recorded against *H. pylori* [126]. Generally, such resistance in *H. pylori* is mainly due to chromosomal mutations, although other factors, such as biofilm formation, the expression of efflux pumps, etc., may also contribute to the development of resistance [129]. The most cases of resistance are recorded against clarithromycin, especially in the Pacific, Mediterranean, and European regions. Resistance against Levoflaxacin shows a similar pattern, with widespread prevalence in the European region, North and South America, and Africa, while Metronidazole shows the highest occurrence of resistance, with as much as 91% cases being resistance, while the lowest number of cases of resistance has been the recorded against Amoxicillin, with less than 1% in most countries [128,130,131]. Following the rising cases of antibiotic resistance and the failure of triple therapy in eradicating the infection, different models of alternative treatment have been proposed in the Maastricht V/Florence consensus report. These include bismuth-based therapy and quadruple therapy [126] (Table 2).

### 5.4. Maastricht VI/Florence Consensus

The Maastricht consensus reports (later, the Maastricht/Florence consensus reports) are a series of guidelines with a prime focus on the management of *H. pylori* infections, mediated by a group of experts from various countries around the world, which represent a global consensus. These guidelines were first established in 1997, and since then, these guidelines have been updated five times, with the most recent guidelines, the Maastricht VI/Florence guidelines, released in 2022. In these editions, the treatment options, diagnoses methods, and management of *H. pylori*-related diseases have been changed, as per the global reports on the success of these factors. The Maastricht VI/Florence report has recommended the administration of triple therapy (clarithromycin, amoxicillin, and a PPI) or a BQT (bismuth, tetracycline, metronidazole, and PPI) as the standard first-line treatment in cases of no-to-low clarithromycin resistance. High-dose PPI-amoxicillin dual therapy in cases of the failure of bismuth quadruple therapy is recommended as a first-line treatment in cases of clarithromycin resistance [132,133]. The report further recommends the use of Potassium-Competitive Acid Blockers (P-CAB) in cases of antimicrobial resistant infections. The other alternatives that have been recommended include a fluoroquinolone-containing quadruple or triple therapy. However, in the regions where fluoroquinolone resistance is predominant, a BQT involving alternative antibiotics or a rescue treatment containing rifabutin is also recommended.

**Table 2 microorganisms-12-00222-t002:** Comparison of the treatment strategies currently available against *H. pylori.*

Treatment Option	Drugs Employed	Duration of Therapy	References
Triple therapy (PPI+ two antibiotics)	PPI, Clarithromycin, Amoxicillin (or Metronidazole)	7 days	[134]
Bismuth Quadruple therapy (BQT)	PPI, bismuth, tetracycline, and metronidazole	14 days	[40]
Levofloxacin-containing triple therapy	PPI, levofloxacin, amoxicillin	14 days	[135]
Levofloxacin-amoxicillin quadruple therapy	PPI, bismuth, levofloxacin, amoxicillin	10 days	[136]
Tetracycline-levofloxacin quadruple therapy	PPI, bismuth, levofloxacin, tetracycline	10 days	[137]
Concomitant therapy (non-bismuth therapy)	PPI, amoxicillin, clarithromycin, and a nitrimidazole		[134]
Sequential therapy (dual)	PPI and amoxicillin for 5 days, followed by triple therapy (PPI, clarithromycin, and tinidazole) for next 5 days	10 days	[138]

## 6. Future Directions

### 6.1. Challenges and Opportunities

Over recent years, significant progress has been made to facilitate our understanding of the pathogenic bacterium that is *H. pylori*. Its role in chronic gastritis and the development of peptic ulcer disease is now well recognized. More importantly, the role of *H. pylori* as a risk factor in the development of gastric cancer has emerged as one of the most sought-after fields of research, with appreciable efforts being made for the timely diagnosis, treatment, and eradication of the disease and the prevention/treatment of the resulting oncogenesis. A meta-analysis conducted in 2023 revealed a reduced rate of occurrence of *H. pylori* infections, resulting in a concomitant decline in the development of peptic ulcer disease, while the prevalence of antibiotic-resistant *H. pylori* is higher than it has been in the last 40 years [139]. Among the newer treatment options being developed are a quadruple therapy comprising of bismuth and the use of probiotics, tailored therapy, phytomedicine, and nanotechnology [140,141].

### 6.2. Advancements in Research

Considering the varied resistance profiles exhibited by *H. pylori,* including single-drug, multidrug, and the previously described heteroresistance, alternative treatment strategies are needed, and improved diagnostic tools are warranted for the optimized selection of treatment choice. The intimate correlation between *H. pylori* infection and the development of gastric cancer provide the potential for an effective vaccination strategy, which may be used to prevent or treat the onset and establishment of gastric cancer in patients infected with the bacterium. However, such vaccination strategies are challenging, owing to the variation in the strains and the complicated mechanisms of the bacteria to evade the immune responses of the host. Surface adhesins, such as BabA, SabA, OipA, etc., have the potential to be employed as biomarkers to help detect the onset of gastric carcinoma, as revealed in a study conducted by Su et al. (2016) [141]. Such biomarkers could also be used as immune system triggers in the form of vaccines to eradicate the infection. Nanotechnology has, in recent years, emerged as an interesting field in biomedical research, providing new avenues for the treatment of various diseases, including *H. pylori*-related diseases. Their targeted drug-delivery potential and the unlimited scope for designability present them as suitable candidates for the development of novel anti-H. *pylori* nanomedicines. Additionally, nanoparticles can also be employed for the accurate and timely diagnosis of the infection, further promoting the efficient management of the disease. Excellent reviews in this regard have already been published [142,143,144]. Similarly, phage therapy, a well-known alternative approach to combat bacterial infections, offers considerable potential to be explored for the effective eradication of *H. pylori* infections [145]. Additionally, the use of probiotics in the treatment of *H. pylori* infection has gained significant traction in recent years. Probiotic bacteria, such as Lactobacilli and Bifidobacterium, have been shown to alleviate infections caused by *H. pylori*, while also improving the immune system function and the performance of antibiotics [146]. Personalized therapy is another novel approach being explored for the effective treatment of *H. pylori* infections. It relies on confirmation of the antibiotic susceptibility of *H. pylori* serotypes extracted from patients after subjecting the bacterial strain to a series of antibiotic susceptibility tests. Additionally, it involves a treatment regime specifically tailored to the individual’s needs, according to their family history, genomic makeup, health history, etc. Such an approach is especially useful considering the steadily rising cases of *H. pylori* resistance against the conventional choice of treatment, i.e., PPI and CAM [147]. While antibiotic therapy has proven to be significantly successful in the management of the infection, the steadily growing cases of antibiotic resistance and resistance against other agents like proton pump inhibitors highlights the need for the development of alternative therapies and, for now, the careful monitoring of antibiotic administration, weighed on a case-by-case basis. Undeniably, antibiotic stewardship has a significant role to play in the effective management of the infection and its wider effects. The judicious and well-regulated use of antibiotics can help minimize the development of resistance cases and still effectively combat the infection, providing a much-needed paradigm shift in the treatment approaches against *H. pylori*. The optimization of the dosage and duration of antibiotics, coupled with susceptibility testing and surveillance programs, is critical for the effective management of the infection [148].

Effective diagnosis calls for the development of new diagnostic procedures, and certain novel tools have recently emerged as an interesting alternative to the currently used methods, including multiplex PCR, fecal microbiota analysis, novel point-of-care testing techniques based on loop-mediated isothermal amplification (LAMP) and PCR detection or rapid antigen tests, confocal laser endomicroscopy (CLE), endoscopic submucosal dissection (ESD), etc. [149]. Although such efforts and emerging technologies seem promising, certain challenges and limitations still remain, including poor patient compliance, the prevailing, intrinsic physiological conditions during treatment, bacterial resistance to antibiotics and other antimicrobials, etc., which must be carefully addressed in order to optimize the diagnosis, treatment and successful eradication of *H. pylori* and further eradicate the risks of complications.

## 7. Conclusions

This review not only consolidates the current knowledge, but also illuminates the potential avenues for future research and clinical interventions. Its relevance to healthcare professionals, researchers, and clinicians is underscored by the emphasis on accuracy in diagnosis, the imperative need for antibiotic stewardship, and the promising horizon of personalized medicine. Additionally, it is not only serves as a valuable reference for those navigating the complexities of *H. pylori,* but also inspires further inquiries into refining the diagnostic and therapeutic approaches in the ongoing pursuit of mitigating the impact of this prevalent and dynamic gastrointestinal pathogen.

## Figures and Tables

**Figure 1 microorganisms-12-00222-f001:**
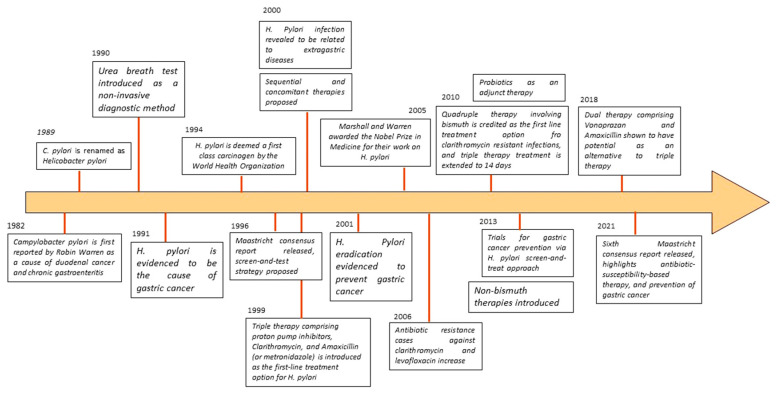
A timeline of significant events related to *H. pylori* discovery and the management of infection.

**Figure 2 microorganisms-12-00222-f002:**
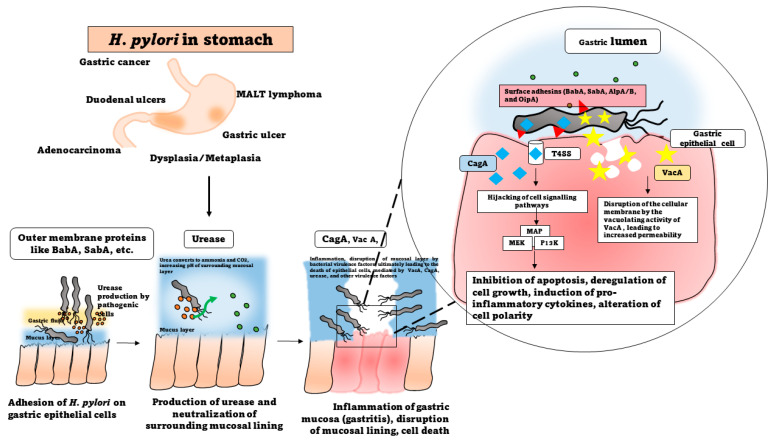
Pathogenesis of *H. pylori.*

## Data Availability

Not applicable.

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
