# Peer review of "Helicobacter pylori: A Contemporary Perspective on Pathogenesis, Diagnosis and Treatment Strategies"

_microorganisms, 2024, doi:10.3390/microorganisms12010222_

Round 1

Reviewer 1 Report

Comments and Suggestions for Authors This review provides an updated and comprehensive overview of the current understanding of H. pylori infection, focusing on its pathogenesis, diagnosis, and treatment strategies and guide both clinicians and researchers toward effective management and future directions in combating Hp. However, in the treatment section, the review is not comprehensive and the references are too outdated. It is now the Maastricht VI/Florence Consensus, and the latest treatment regimens are not mentioned, such as the use of P-CAB and high-dose Dual therapy.

Comments on the Quality of English Language

Minor editing of English language required

Author Response

Response to referees' comments

Manuscript title: “Helicobacter pylori: A contemporary perspective on pathogenesis, diagnosis, and treatment strategies”.

Manuscript IDmicroorganisms-2800088

************************************************************************************************

I would like to express my sincere appreciation to both the reviewer and the editor for their meticulous examination of my article. Their insightful comments and constructive feedback have significantly enriched the content and quality of the manuscript. Their expertise and attention to detail have not only strengthened the scientific merit of the work but also enhanced its overall clarity. I am grateful for their time, dedication, and valuable contributions, which have undoubtedly played a crucial role in refining the article and elevating its scholarly impact.

Reviewer 1

Query 1:

This review provides an updated and comprehensive overview of the current understanding of H. pylori infection, focusing on its pathogenesis, diagnosis, and treatment strategies and guiding both clinicians and researchers toward effective management and future directions in combating Hp. However, in the treatment section, the review is not comprehensive, and the references are too outdated. It is now the Maastricht VI/Florence Consensus, and the latest treatment regimens are not mentioned, such as the use of P-CAB and high-dose Dual therapy.

Response:

I wish to convey my gratitude for the valuable suggestion offered. The Maastricht VI/Florence consciences have been integrated into the designated section heading 5.4 (lines 463-482). Additionally, in accordance with the esteemed reviewer's suggestion, the latest references (like 8-16, 20, 21, 44-50, 113,116,119,124 etc) have been appropriately cited.

Query 2:

Minor editing of the English language required

Response:

In light of the reviewers' observations, we have diligently examined the manuscript and systematically revised the English language where deemed necessary.

Reviewer 2 Report

Comments and Suggestions for Authors

In general, the review contains valuable current information on the selected topic.

Despite this, adding some information may be recommended:

1. Add brief global statistical data on H. pylori infection and related diseases.

2. Add information on ways of spreading infection and routes of infection.

3. Add information on H. pylori resistance to low pH (mechanisms, surviving at low pH, etc.).

Recommended editorial changes:

1. Italicize species names H. pylori thought the text.

2. Elements (including all captions) in Figure 2 should be enlarged.

Comments on the Quality of English Language

The text should be checked for typos.

Author Response

Response to referees' comments

Manuscript title: “Helicobacter pylori: A contemporary perspective on pathogenesis, diagnosis, and treatment strategies”.

Manuscript IDmicroorganisms-2800088

************************************************************************************************

I would like to express my sincere appreciation to both the reviewer and the editor for their meticulous examination of my article. Their insightful comments and constructive feedback have significantly enriched the content and quality of the manuscript. Their expertise and attention to detail have not only strengthened the scientific merit of the work but also enhanced its overall clarity. I am grateful for their time, dedication, and valuable contributions, which have undoubtedly played a crucial role in refining the article and elevating its scholarly impact.

Reviewer 2

In general, the review contains valuable current information on the selected topic.

Despite this, adding some information may be recommended:

Query 1:

Add brief global statistical data on H. pylori infection and related diseases.

Response:

Pursuant to the reviewer's recommendation, we have integrated the global scenario of H. pylori infection into the introduction section, delineated between lines 59 and 74.

Query 2:

Add information on ways of spreading infection and routes of infection.

Response:

We express our gratitude for the invaluable comment provided by the reviewer. The dissemination and route of infection of H. pylori have been integrated under the designated section heading 2.1, specifically within lines 105-115.

Query 3:

Add information on H. pylori resistance to low pH (mechanisms, surviving at low pH, etc.).

Response:

We express our gratitude for the insightful suggestion. Subsequent to this recommendation, we have included pertinent information concerning the survival of H. pylori at low pH, strategically placed under the subheading 2.2.4 (lines 183-196). within the manuscript.

Recommended editorial changes:

  1. Italicize species names H. pylori through the text.

Response: The species name is italicized consistently throughout the manuscript.

  1. Elements (including all captions) in Figure 2 should be enlarged.

Response:  The figure is enlarged and arranged appropriately.

Round 2

Reviewer 1 Report

Comments and Suggestions for Authors this article offers valuable insights into the contemporary knowledge of Helicobacter pylori infection, guiding both clinicians  and researchers toward effective management and future directions in combating this global health challenge.